# Maternal and child outcomes for pregnant women with pre-existing multiple long-term conditions: protocol for an observational study in the UK

Siang Ing Lee [ID],[1] Holly Hope [ID],[2] Dermot O'Reilly [ID],[3] Lisa Kent [ID],[3] Gillian Santorelli [ID],[4] Anuradhaa Subramanian [ID],[1] Ngawai Moss,[5] Amaya Azcoaga-Lorenzo [ID],[6,7] Adeniyi Francis Fagbamigbe [ID],[6,8] Catherine Nelson-Piercy [ID],[9] Christopher Yau [ID],[10,11] Colin McCowan [ID],[6] Jonathan Ian Kennedy [ID],[12] Katherine Phillips [ID],[1] Megha Singh [ID],[1] Mohamed Mhereeg [ID],[12] Neil Cockburn [ID],[1] Peter Brocklehurst [ID],[1] Rachel Plachcinski [ID],[5] Richard D Riley [ID],[1] Shakila Thangaratinam [ID],[13,14] Sinead Brophy [ID],[12] Sudasing Pathirannehelage Buddhika Hemali Sudasinghe,[1] Utkarsh Agrawal [ID],[6] Zoe Vowles [ID],[9] Kathryn Mary Abel [ID],[2,15] Krishnarajah Nirantharakumar [ID],[1] Mairead Black [ID],[16] Kelly-Ann Eastwood [ID],[3,17] on behalf of MuM-PreDiCT

KN, MB and K-AE are joint senior authors.

**Correspondence to**
Professor Krishnarajah Nirantharakumar;
k.niranthan@bham.ac.uk

## ABSTRACT

**Introduction** One in five pregnant women has multiple pre-existing long-term conditions in the UK. Studies have shown that maternal multiple long-term conditions are associated with adverse outcomes. This observational study aims to compare maternal and child outcomes for pregnant women with multiple long-term conditions to those without multiple long-term conditions (0 or 1 long-term conditions).

**Methods and analysis** Pregnant women aged 15–49 years old with a conception date between 2000 and 2019 in the UK will be included with follow-up till 2019. The data source will be routine health records from all four UK nations (Clinical Practice Research Datalink (England), Secure Anonymised Information Linkage (Wales), Scotland routine health records and Northern Ireland Maternity System) and the Born in Bradford birth cohort. The exposure of two or more pre-existing, long-term physical or mental health conditions will be defined from a list of health conditions predetermined by women and clinicians. The association of maternal multiple long-term conditions with (a) antenatal, (b) peripartum, (c) postnatal and long-term and (d) mental health outcomes, for both women and their children will be examined. Outcomes of interest will be guided by a core outcome set. Comparisons will be made between pregnant women with and without multiple long-term conditions using modified Poisson and Cox regression. Generalised estimating equation will account for the clustering effect of women who had more than one pregnancy episode. Where appropriate, multiple imputation with chained equation will be used for missing data. Federated analysis will be conducted for each dataset and results will be pooled using random-effects meta-analyses.

## STRENGTHS AND LIMITATIONS OF THIS STUDY

⇒ The study will use rich data sources from routine health records from all four UK nations and a birth cohort.
⇒ Beyond examining maternal outcomes, linked mother–baby data and the birth cohort data will allow for the exploration of children's outcomes.
⇒ Key limitations include missing data, misclassification bias due to inaccurate clinical coding and residual confounding.

**Ethics and dissemination** Approval has been obtained from the respective data sources in each UK nation. Study findings will be submitted for publications in peer-reviewed journals and presented at key conferences.

## INTRODUCTION

Maternal single long-term conditions such as cardiac conditions, chronic kidney disease and epilepsy are associated with adverse pregnancy outcomes.[1–4] This is likely to be compounded when the pregnant woman has two or more long-term physical or mental health conditions (multimorbidity). Some conditions may need different treatments from different healthcare teams, thereby increasing the treatment burden and complexity of care.[5] Recent evidence has shown that maternal multiple long-term conditions are associated with adverse outcomes for women and their children, such as severe maternal morbidity

and mortality, pre-eclampsia, emergency caesarean birth, preterm birth and low birth weight.[6–8] In the UK 2016–2018 national maternal mortality report, 90% of women who died during or up to a year after pregnancy had multiple health or social problems.[9]

Currently, one in five pregnant women has multiple long-term conditions prior to pregnancy in the UK.[10] The number of pregnant women with pre-existing multiple long-term conditions is likely to increase as women are getting pregnant later in life and with higher body weight.[11–14] As this becomes an increasingly important issue, information on pregnancy, maternal and child outcomes is crucial for women and their healthcare professionals to make informed decisions on preconception and pregnancy care planning. However, there remains a lack of evidence to guide care pathways for pregnant women with multiple long-term conditions.[8 15]

Healthcare is free in the UK and over 98% of the population are registered at a general practice (akin to family practice in other countries).[16] General practices not only provide primary and community healthcare, but they also serve as the main point of contact for referrals to specialist clinical services and provide the majority of prescribing outside of a hospital setting.[16] In the UK, pregnant women are recommended to have their booking appointment before 10 weeks gestation.[17] This is the pregnant woman's first midwife or doctor appointment, where they undergo health and social care assessment of needs and risks for her pregnancy.[18] Over 97% of births occur in healthcare settings in England and Wales.[19] Therefore, routine health records in primary and secondary care in the UK offer a rich data source for observational studies of pregnant women and their children.

This observational study aims to compare outcomes for women with multiple long-term conditions to those without multiple long-term conditions. Outcomes studied will include those for women and their children. Datasets from routine health records from all four UK nations (England, Wales, Scotland and Northern Ireland) will be used. In addition, the Born in Bradford birth cohort from a deprived, ethnically diverse city in the UK, will also be used.[20]

The four research objectives are to examine the association between maternal pre-existing multiple long-term conditions with: (a) antenatal, (b) peripartum, (c) postnatal and long-term outcomes and (d) mental health outcomes. The findings from each research objective will be published in a separate paper.

## METHODS AND ANALYSIS
### Study design
This is a cohort observational study using data from routine healthcare records and a birth cohort in the UK.

### Study population and eligibility criteria
The study population will consist of women aged 15–49 years old at conception, with pregnancies beginning between 1 January 2000 and 31 December 2019 in the UK. Date of conception (pregnancy start date) will be defined as the first day of the last menstrual period or gestational day 0. To ensure sufficient quality data, eligible women must have health records that meet the standard data quality checks as defined by each data source and 1 year's worth of health records prior to index pregnancy.

### Data sources
Table 1 presents the five data sources that will be used. Each UK devolved nation is represented by a population based routine health record dataset, with good national coverage for Wales, Scotland and Northern Ireland and a representative sample for England.[16] The exposure status will be determined from primary care records for Clinical Practice Research Datalink (CPRD) and Secure Anonymised Information Linkage (SAIL), with CPRD GOLD representing 5% of UK general practices,[21] and SAIL covering 80% of Welsh general practices.[22] For Scotland's linked routine records and Northern Ireland Maternity System (NIMATS), the exposure status will be determined from hospital and prescribing records.

CPRD and SAIL's primary care data offer the opportunity to study outcomes that may not be captured in secondary care. For instance, vomiting in pregnancy, miscarriage and neurodevelopmental conditions in children. The Scottish dataset provides detailed information on the different types of hospital attendances, including psychiatric admissions and accident and emergency attendances. NIMATS's unique first antenatal visit dataset is a good source of pre-pregnancy clinical data not available in other datasets.

As routine health records were not collected for research purposes, it is prone to missing data. Therefore, we have also included Born in Bradford, a regional birth cohort (2007–2011) where data were collected systematically and longitudinally from pregnancy, childhood through to adult life.

### Exposure
The exposed group will consist of pregnant women with multiple long-term conditions. Measurements of multiple long-term conditions are variable in existing literature.[23 24] Currently only Bateman *et al*'s Maternal Comorbidity Index has been developed specifically for obstetric research.[25 26] It consists of 20 health conditions and included conditions arising in pregnancy such as gestational hypertension, pre-eclampsia and placenta praevia.[26] This limits the ability to study the impact of pre-existing long-term conditions on maternal and child health and the implication for long-term condition management preconception.[8]

In this study, we shall define multiple long-term conditions as two or more long-term physical or mental health conditions that pre-existed before pregnancy. Pregnancy related complications will not be included as they will be studied as outcomes. Multiple long-term conditions will be defined from a list of 79 health conditions previously

**Table 1** Summary of data sources

| Name of data source | Country | Population: pregnant women | Exposure: maternal multiple long-term conditions status | Outcomes: pregnant women | Outcomes: children |
|---|---|---|---|---|---|
| Clinical Practice Research Datalink (CPRD)[16] | England | Pregnancy register (primary care) | Primary care routine health records | Primary care records, hospital admissions, death registration | Mother–baby linked data: primary care records, hospital admissions, death registration |
| Secure Anonymised Information Linkage (SAIL)[22] | Wales | Births from National Community Child Health Dataset | Primary care routine health records | Primary care records, hospital admissions, death registration | Mother–baby linked data: primary care records, hospital admissions, death registration |
| Scotland routine health records | Scotland | Scottish Maternity Records, pregnancy-related hospital admissions | Hospital admissions, psychiatric admissions, accident and emergency attendances, prescriptions | Hospital admissions, psychiatric admissions, accident and emergency attendances, death registration | Mother–baby linked data: hospital admissions, psychiatric admissions, accident and emergency attendances, death registration |
| Northern Ireland Maternity System (NIMATS)[43] | Northern Ireland | Maternity booking (first antenatal) appointment records, birth related hospital admissions | Maternity booking (first antenatal) appointment records, birth related hospital admissions, prescriptions | Hospital admissions | Mother–baby linked data: hospital admissions |
| Born in Bradford[44] | Bradford, England | Birth cohort of over 13 500 children born from around 12 500 mothers at the Bradford Royal Infirmary between March 2007 and June 2011 | Primary care routine health records | Data from birth cohort: clinical data Data from linked health records: maternity, primary care, hospital admissions | Data from birth cohort: offspring developmental, clinical and education data Data from linked health records: primary care, hospital admissions |

described in our epidemiological work (Box 1) and will be measured with simple count.[10] This list was compiled from existing multimorbidity literature[9 24 27] and a workshop with our multidisciplinary research advisory group, including patient representatives and clinicians.[10] Selection of health conditions were based on: (a) prevalence; (b) potential to impact on pregnancy outcomes; (c) considered important by women and (d) recorded in the study datasets.[10] The phenome definitions for these health conditions have previously been described in our epidemiological work.[10] For health conditions that are transient and episodic in nature (eg, asthma, eczema, depression and anxiety), we will only include the condition if it is active, which we have defined as requiring a doctors' consultation or medical prescription in the 12 months preceding pregnancy.[10] Sensitivity analysis will be performed defining maternal multiple long-term conditions with a different list of health conditions by D'Arcy and Knight.[28]

Exposure will be ascertained by the presence of diagnostic or prescriptions codes, including Read (to identify exposures in primary care data) and International Classification of Disease 10th version (ICD-10, secondary care).

## Comparator
### Multiple long-term conditions versus no multiple long-term conditions
Comparisons will be made with the following exposure group:
► Pregnant women with multiple long-term conditions.
► Pregnant women with increasing counts of long-term health conditions.

► Pregnant women with different combinations of long-term health conditions.
► Pregnant women in different health condition clusters (identified from ongoing clustering analyses).

The selection of which combinations and clusters of long-term conditions to study will be based on how common they are and their clinical relevance, following consultation with patient representatives and clinicians in our research team. Pregnant women with no multiple long-term conditions (ie, no or single long-term conditions) will be the common comparator group.

### Multiple long-term conditions with and without mental illness
In addition, we will also compare the outcomes for pregnant women who have mental health conditions as part of their multiple long-term conditions against pregnant women with multiple long-term conditions who do not have mental health conditions.

## Outcomes
The outcomes will be grouped into the following four categories based on the research objectives: (a) antenatal, (b) peripartum, (c) postnatal and long-term outcomes and (d) mental health outcomes. Examples of outcomes are provided as follows, based on existing core outcome sets for pregnancy and childbirth.[29 30] The definitive list of outcomes will be confirmed once the development work for a core outcome set for studies of pregnant women with multiple long-term conditions is completed.[31] Outcomes will be ascertained from the study datasets (1 January 2000 to 31 December 2019) using clinical codes, such as

## Box 1 List of 79 health conditions defining multiple long-term conditions in pregnancy

### Cancers
1. All cancers
⇒ Solid cancers
⇒ Haematological cancers
⇒ Metastatic cancers
⇒ Exclude basal cell carcinoma

### Cardiovascular disease
2. Hypertension
3. Ischaemic heart disease and myocardial infarction
4. Heart failure
5. Stroke
⇒ Transient ischaemic attack
⇒ Ischaemic stroke
⇒ Haemorrhagic stroke
⇒ Unspecified stroke
6. Atrial fibrillation
7. Congenital heart disease
8. Valvular heart disease (mitral, aortic, mixed)
9. Cardiomyopathy

### Dermatology
10. Eczema
11. Psoriasis
12. Autoimmune skin disease
⇒ Vitiligo
⇒ Alopecia areata
13. Other dermatological conditions
⇒ Seborrheic dermatitis
⇒ Rosacea
⇒ Hidradenitis suppurativa
⇒ Lichen planus

### Ear, nose, throat
14. Profound deafness
15. Allergic rhinitis and allergic conjunctivitis

### Eye
16. Inflammatory eye disease
⇒ Scleritis and episcleritis
⇒ Anterior uveitis
⇒ Posterior uveitis
17. Cataract
18. Diabetic eye disease
19. Severe blindness
20. Retinal detachment

### Gastroenterology
21. Irritable bowel syndrome
22. Inflammatory bowel disease
⇒ Ulcerative colitis
⇒ Crohn's disease
23. Coeliac disease
24. Chronic liver disease
⇒ Chronic hepatitis B and C
⇒ Alcoholic liver disease
⇒ Autoimmune liver disease
⇒ Cirrhosis
⇒ Non-alcoholic fatty liver disease
25. Peptic ulcer

## Box 1 Continued

26. Gall stones

### Gynaecology
27. Polycystic ovarian syndrome
28. Endometriosis
29. Fibroids
30. Infertility

### Haematology
31. History of venous thromboembolism
⇒ Deep vein thrombosis
⇒ Pulmonary embolism
32. Primary thrombocytopenia
33. Haemophilia
34. Sickle cell anaemia
35. Pernicious anaemia

### Mental health
36. Depression
37. Anxiety
⇒ Panic disorder
⇒ Phobia disorder
⇒ Post-traumatic stress disorder
38. Severe mental illness
⇒ Bipolar affective disorder
⇒ Schizophrenia
⇒ Psychosis
39. Eating disorder
40. History of alcohol use disorder (misuse/dependence)
41. History of substance misuse
42. Others
⇒ Obsessive compulsive disorder
⇒ Self-harm
⇒ Personality disorder
⇒ Dissociative disorder

### Neurodevelopmental conditions
43. Neurodevelopmental conditions
⇒ Learning disability
⇒ Attention deficit hyperactivity disorder
⇒ Autistic spectrum disorder

### Rheumatology
44. Systemic lupus erythematosus
45. Spondylarthritis
⇒ Psoriatic arthritis
⇒ Ankylosing spondylitis
46. Inflammatory arthritis
⇒ Rheumatoid arthritis
⇒ Sjogern's syndrome
⇒ Raynaud's syndrome
⇒ Systemic sclerosis
⇒ Primary systemic vasculitis
47. Ehler's Danlos Syndrome (EDS) type 3 (hypermobile EDS)

### Orthopaedic
48. Scoliosis
49. Vertebral disorder
⇒ Intervertebral disc disorder
⇒ Spondylosis
⇒ Spondylolisthesis
⇒ Collapsed vertebrae

**Box 1  Continued**

⇒ Spinal stenosis
50. Chronic back pain
51. Osteoporosis
52. Osteoarthritis

**Neurology**
53. Migraine
54. Other chronic headache (including cluster headache, tension headache)
55. Epilepsy
56. Multiple sclerosis
57. Spina bifida
58. Idiopathic intracranial hypertension
59. Peripheral neuropathy
60. Other neurological conditions/musculoskeletal disorders
⇒ Chronic fatigue syndrome/myalgic encephalomyelitis
⇒ Fibromyalgia
⇒ Chronic pain syndrome (includes chronic regional pain syndrome, myofascial pain syndrome)

**Respiratory**
61. Asthma
62. Chronic obstructive pulmonary disease
63. Obstructive sleep apnoea
64. Pulmonary fibrosis, interstitial lung disease
65. Pulmonary hypertension
66. Bronchiectasis
67. Cystic fibrosis
68. Sarcoidosis

**Renal**
69. Chronic kidney disease
70. Urinary tract stones

**Endocrine**
71. Diabetes mellitus
72. Thyroid disorder
73. Pituitary disorder
74. Adrenal benign tumour
75. Hyperparathyroidism

**Other**
76. HIV infection/AIDS
77. Turner's syndrome
78. Marfan's syndrome
79. Solid organ transplant

Read, ICD-10 and Operating Procedures Codes Classification of Interventions and Procedures.

### Antenatal

Antenatal outcomes occur from conception to before the onset of childbirth. Examples for women include miscarriage, gestational hypertension, pre-eclampsia, gestational diabetes, venous thromboembolism, placenta abruption and antenatal hospital admissions. Examples for children include fetal growth restriction.

### Peripartum

Peripartum outcomes occur during and immediately after childbirth. This category will also include survival outcomes for women and children. Examples for women include mode of birth (spontaneous vaginal birth, birth with forceps/ventouse, caesarean birth), postpartum haemorrhage, severe maternal morbidity, admission to intensive care and maternal death. Examples for children include preterm birth, small for gestational age, admission to neonatal unit, stillbirth, perinatal death and neonatal death.

### Postnatal and long-term

Postnatal outcomes occur in the 42 days after birth,[32] while long-term outcomes are beyond the peripartum and postpartum period. For women this would include functional outcomes such as incontinence. For children, we will use mother–baby linked primary and secondary care data to study postnatal and long-term outcomes such as congenital anomalies, neurodevelopmental conditions (eg, autism, attention deficit hyperactive disorder and learning difficulty), cerebral palsy and chronic lung disease. The length of follow-up will depend on the availability of data in the routine health records. For example, CPRD has a median follow-up of 5 years.[16] We will also examine postpartum readmission for mother and child.

### Mental health

Mental health outcomes cover the antenatal and postnatal period and will be considered up to 12 months after birth. This is to account for possible delay in women presenting to clinicians and reaching a formal diagnosis. We will consider both: (a) incident and (b) recurrent mental health outcomes, where incident means a woman enters the analysis with no prior record of the specific mental health outcome. A perinatal mental health event is indicated by a primary care visit or hospital admission and includes mental health outcomes of concern in the antenatal and postnatal period (eg, depression, psychosis, post-traumatic stress disorder, self-harm and suicide attempts). Comparing the mental health event rates of pregnant women who have and have not got mental health conditions as part of their multiple long-term conditions will allow us to delineate the contribution of mental and physical morbidity to perinatal mental health outcomes. Children's mental ill health will also be considered (eg, depression and anxiety).

### Covariates

Analyses will adjust for the following covariates. Additional covariates may be added for individual outcomes based on the literature. For example, in analyses of mental health outcomes there will be additional covariates. For the mother, we will include history of any mental illness, for the child we will include maternal history of any mental and/or neurodevelopmental conditions.

Where data for antenatal exposures are available (eg, from NIMATS and Born in Bradford's booking appointments), additional analyses may be conducted where appropriate.

## Maternal age

We shall explore whether the association between maternal age and the outcomes are linear. Where this is not the case and to aid clinical interpretability, we will categorise maternal age at conception into 5-yearly age bands.

## Parity/gravidity

The variable used will depend on availability in study datasets. Where both variables are available, both will be reported with preference given to *parity* (the number of times a woman gave birth at gestation ≥24 weeks); and sensitivity analysis will be conducted using *gravidity* (the number of times a woman has been pregnant).

## Ethnicity

Maternal ethnicity will be categorised based on the variables available and to allow for harmonisation across the datasets: Asian, black, mixed, other and white. Where data permits, we may use more granular categories of ethnicity. Where numbers are too small and risk identifying individuals, such as in NIMATS, we may collapse the categories to white and non-white.

## Social deprivation

The patient level Index of Multiple Deprivation specific to each nation will be used and categorised into quintiles.

## Body mass index

We shall include the latest available pre-pregnancy body mass index for the pregnant women. Where booking data is available before 16 weeks gestation, this will be used (eg, in NIMATS). Body mass index will be considered a covariate instead of a health condition. The WHO's classification of obesity will be used to categorise body mass index: <18.5 kg/m², 18.5 to 24.9 kg/m², 25.0 to 29.9 kg/m², 30.0 to 34.9 kg/m², 35.0 to 39.9 kg/m² and 40+ kg/m².[33] Categories may be combined where numbers are too small.

## Smoking

We shall include the latest available pre-pregnancy smoking status for the pregnant women. Smoking status will be categorised as: non-smoker, ex-smoker and smoker.

## Year (pregnancy start date)

Data quality and clinical guidelines may vary by year. Its effect on outcomes will be accounted for by adjusting for year of conception in the analysis.

## Statistical analysis

We anticipate analyses will commence in June 2023 with study completion by June 2024. Baseline characteristics of the study population and outcomes will be described with summary statistics. Modified Poisson regression will be performed to estimate the relative risks of study outcomes. Cox regression will be performed for longer-term outcomes. The unit of analysis will be the pregnancy episode.

A federated analysis approach will be used as data governance arrangements do not allow pooling of the data across the four nations. Each dataset will be analysed separately following a common study protocol. A common data model will be established and implemented across the dataset, building on our previous work harmonising the phenome definitions for exposure conditions.[10] The effect sizes will be pooled using random-effects meta-analyses with inverse variance weighting for the primary care and secondary care datasets, respectively.[34]

Where rare combinations of health conditions and outcomes may lead to identification of an individual or at the prespecified minimum count allowed by each data source, we will suppress the output.

## Pregnant women with more than one pregnancy episode

An individual may have more than one pregnancy over the study period. The pregnancy episodes of the same woman will not be independent of each other. The severity of the exposure variable (pre-existing multiple long-term conditions) may increase in later pregnancy episodes as the pregnant women accumulates more long-term health conditions. If a woman had an adverse pregnancy outcome, she is more at risk of the same adverse outcome in subsequent pregnancy episodes. We shall account for this clustering effect of women with more than one pregnancy episode during the study period using the Generalised Estimating Equation in the regression analyses.

## Multiple pregnancies

The main analysis will be limited to singleton pregnancies. Outcomes for pregnant women with multiple long-term conditions and multiple pregnancies (ie, twins and higher order pregnancies) will be analysed as a separate cohort.

## Missing data

Where exposure and outcome conditions are identified based on diagnostic codes, the absence of the code will be considered as an absence of the condition. The level and types of missingness of covariates will be reviewed and where appropriate will be addressed with representing missing data as a separate category or multiple imputation with chain equation (MICE). For variables required to compute an outcome, missing values will be imputed using MICE. Example of these variables include birth weight, gestational age and baby's sex to determine preterm birth and small for gestational age. For each outcome, the statistical analyses will be performed on the imputed datasets and the estimates will be pooled with Rubin's rule.

## Sensitivity analyses

We shall conduct sensitivity analyses using (a) complete case analysis, (b) varying definitions of maternal multiple long-term conditions exposure using D'Arcy and Knight's core exposure set[28] and (c) in primiparous women. The latter is to account for the fact that some long-term conditions can arise from complications from a previous pregnancy.

## Patient and public involvement

The research question was informed by discussions with our patient and public involvement (PPI) advisory group and our PPI coinvestigators (NM and RP).

The selection of outcomes is guided by our ongoing work developing a core outcome set for studies of pregnant women with multiple long-term conditions, where patients are key stakeholders.[31]

Our PPI advisory group and PPI coinvestigators will be involved in interpreting the study findings, producing lay summaries and infographics, and disseminating the study findings through their network.

## ETHICS AND DISSEMINATION
### Ethics approval

*CPRD*: CPRD has broad National Research Ethics Service Committee ethics approval for purely observational research using the primary care data and established data linkages. The study has been reviewed and approved by CPRD's Independent Scientific Advisory Committee (reference: 20_181R).

*SAIL*: In accordance with UK Health Research Authority guidance, ethical approval is not mandatory for studies using only anonymised data. The study has been approved by SAIL Information Governance Review Panel.

*Scotland dataset*: The study has been approved by the National Health Service Scotland Public Benefit and Privacy Panel for Health and Social Care (HSC-PBPP) and The University Teaching and Research Ethics Committee (UTREC) from the University of St Andrews.

*NIMATS*: The study has been approved by the Honest Broker Service Governance Board.

*Born in Bradford*: Ethics approval was granted by Bradford National Health Service Research Ethics Committee (ref 07/H1302/112) for the Born in Bradford cohort.

The proposed study is purely observational and will use anonymised research data. The study will not involve participant recruitment. Therefore, consent to participate is not required.

### Consent for publication

This is not applicable as the manuscript is a study protocol. In the proposed study, we will use deidentified study data, therefore consent for publication will not be required.

### Dissemination

Study findings will be submitted for publications in peer-reviewed journals and presented at key conferences for health and social care professionals involved in the care of pregnant women with multiple long-term conditions and their children. We will also organise dissemination events to share our findings with the public, service users, clinicians and researchers.

## DISCUSSION

MuM-PreDiCT is a consortium across all four nations of the UK studying multiple long-term conditions in pregnancy. As part of MuM-PreDiCT's programme of work, we outlined the protocol for an observational study of maternal and child outcomes for pregnant women with multiple long-term conditions, using routine health records and a birth cohort in the UK.

### Comparison with current literature

A recent systematic review found seven observational studies on the association of pre-pregnancy multiple long-term conditions with adverse maternal outcomes.[8] The review found that pre-pregnancy multiple long-term conditions were associated with severe maternal morbidity, hypertensive disorders of pregnancy and acute healthcare use in the perinatal period.[8] Most studies were conducted in the USA.[8] Authors of the review commented that many studies included conditions arising in pregnancy in defining multiple long-term conditions, making it difficult to examine the impact of chronic conditions on maternal health.[8]

This proposed study will be based in the UK and will use a broad range of long-term conditions selected by women and clinicians to define multiple long-term conditions. Pregnancy related conditions and complications will be treated as study outcomes and will not be included in the exposure's definition. We will also study outcomes across all stages of pregnancy and outcomes for both women and their children.

### Strengths and limitations

This proposed study will use routine health records from all four nations of the UK (England, Scotland, Wales and Northern Ireland). The available data sources consist of anonymised patient records from primary and secondary care, community prescription data, and maternity care data from routine booking appointments (first antenatal appointment offered universally and as the gateway to access maternity care in the UK).

Rich data will also be available from a birth cohort from Bradford, an ethnically diverse population in England. Beyond examining maternal outcomes, linked mother–baby data and the birth cohort data will allow for the exploration of child outcomes. The key strength of this proposed study therefore is the generalisability of study findings to the UK population. Observing similar effect sizes across the different datasets will also increase the confidence in the study findings. Conversely, discrepancy in findings will stimulate further exploration of the datasets which may generate new knowledge.

As this is an observational study using anonymised routine health records, key limitations include missing data, misclassification bias due to inaccurate clinical coding and residual confounding.

Maternal multimorbidity will be quantified with simple counts. A systematic review of comorbidity indices used in maternal health research found three indices: Maternal

Comorbidity Index, Charlson comorbidity index and Elixhauser comorbidity index.[25] Only the Maternal Comorbidity Index was developed from pregnant and postpartum women.[25] It was developed using hospital data with 20 maternal comorbidities but it included pregnancy related complications and factors such as multiple gestation, gestational diabetes and hypertension disorder of pregnancy.[25 26] In contrast, the list of health conditions we will use to define maternal pre-existing multimorbidity is more comprehensive and included leading causes of indirect maternal death (eg, epilepsy) and mental health conditions.

Nevertheless, when using simple counts to quantify multiple long-term conditions, the severity of each health conditions will not be captured. The dose–response relationship will only be reflected in the total number of pre-existing long-term conditions. For example, we will not be able to distinguish the outcomes for a pregnant woman with diet controlled diabetes and mild asthma from a pregnant woman with insulin dependent diabetes and brittle asthma. However, pregnant women with severe conditions are more likely to receive intense specialist care than pregnant women with mild conditions. As the number of pregnant women with greater disease severity is likely to be smaller than those with milder condition, adverse pregnancy outcomes may be underestimated.

Exposure and outcome events are only captured in routine health records when the pregnant women have presented to primary or secondary care and therefore the true prevalence and incidence may be underestimated. Health conditions that are managed conservatively in primary care, such as depression, anxiety and miscarriage, may not be captured in secondary care datasets. Events such as termination of pregnancy that occurred outside of the traditional healthcare settings may also be underestimated.[35] Similarly, antenatal hospital admission data may not reflect the full burden of additional antenatal appointments or acute care attendances, as care accessed through other routes may not be captured.

Body mass index, which encompasses underweight and obese categories, will be studied as a covariate instead of being counted as part of multimorbidity. There is much debate around whether obesity should be considered a disease[36] or a risk factor for other long-term conditions such as cardiometabolic conditions and cancers.[37–39] What is clear is pre-pregnancy maternal obesity is associated with adverse pregnancy outcome and dedicated care guideline has been established to manage this risk.[40 41]

### Clinical implications

Current obstetric guidelines for pregnant women with medical conditions are focused on specific and single health conditions. There are currently no guidelines for the management of pregnant women with multiple long-term conditions in the UK. The heterogeneity of multiple long-term conditions means an all-encompassing guideline for every possible combination of long-term conditions would not be possible. Indeed the English national guideline for multimorbidity focuses on general approaches such as coordinated and holistic care, improving quality of life by reducing treatment burden and shared decision making between patients and clinicians.[42] A guideline for multiple long-term conditions (multimorbidity) in pregnancy is likely to follow the same principles but with additional focus on the maternity care aspect.

The basis of shared decision making is the provision of evidence based information. As observed in the systematic review, there is currently a lack of evidence on the consequences of pregnancy for women with multiple long-term conditions.[8] Our PPI advisory group and preliminary findings from our core outcome set development work have highlighted how women valued having information to help them mentally prepare to face potential adverse pregnancy outcomes. The output from this study will therefore provide valuable information for women to make informed decision with their clinicians about family planning and their preconception, pregnancy and postpartum care. It will also provide valuable information to guide the future design of care pathway for women with multiple long-term conditions.

**Author affiliations**
[1]Institute of Applied Health Research, University of Birmingham, Birmingham, UK
[2]Centre for Women's Mental Health, Faculty of Biology Medicine & Health, The University of Manchester, Manchester, UK
[3]Centre for Public Health, Queen's University Belfast, Belfast, UK
[4]Bradford Institute for Health Research, Bradford Teaching Hospitals NHS Trust, Bradford, UK
[5]Patient and Public Representative, London, UK
[6]Division of Population and Behavioural Sciences, University of St Andrews School of Medicine, St Andrews, UK
[7]Instituto de Investigación Sanitaria Fundación Jimenez Diaz, Hospital Rey Juan Carlos, Mostoles, Spain
[8]Department of Epidemiology and Medical Statistics, University of Ibadan College of Medicine, Ibadan, Nigeria
[9]Guy's and St Thomas' NHS Foundation Trust, London, UK
[10]Nuffield Department of Women's and Reproductive Health, University of Oxford, Oxford, UK
[11]Health Data Research UK, London, UK
[12]Data Science, Medical School, Swansea University, Swansea, UK
[13]WHO Collaborating Centre for Global Women's Health, University of Birmingham Institute of Metabolism and Systems Research, Birmingham, UK
[14]Department of Obstetrics and Gynaecology, Birmingham Women's and Children's NHS Foundation Trust, Birmingham, UK
[15]Greater Manchester Mental Health NHS Foundation Trust, Manchester, UK
[16]Aberdeen Centre for Women's Health Research, University of Aberdeen School of Medicine Medical Sciences and Nutrition, Aberdeen, UK
[17]University Hospitals Bristol and Weston NHS Foundation Trust, Bristol, UK

**Twitter** Siang Ing Lee @IngLee17, Holly Hope @hfhope, Lisa Kent @LisaKent_QUB, Anuradhaa Subramanian @anuradhaa_s, Ngawai Moss @ngawai_n, Amaya Azcoaga-Lorenzo @amaya_azcoaga, Adeniyi Francis Fagbamigbe @franstel74, Catherine Nelson-Piercy @nelson_piercy, Christopher Yau @cwcyau, Megha Singh @meghasingh_16, Neil Cockburn @defnatlynotneil, Peter Brocklehurst @brocklehurst_p, Rachel Plachcinski @Stroppybrunette, Richard D Riley @ Richard_D_Riley, Shakila Thangaratinam @thangaratinam, Sinead Brophy @@ SineadBr, Utkarsh Agrawal @utkarsh30591, Zoe Vowles @zoe_vowles, Krishnarajah Nirantharakumar @Nirantharakumar, Mairead Black @maireadblack and Kelly-Ann Eastwood @K_AEastwood

**Collaborators** MuM-PreDiCT Group: Francesca Crowe, Beck Taylor, Charles Gadd, Helen Dolk, Maria Loane, Louise Locock, Sharon McCann, Muhammad Usman,

Christine Damase-Michel, Stephanie Hanley, Jingya Wang, Steven Wambua, Natalia Hong, Astha Anand.

**Contributors** SIL: conceptualisation, funding acquisition, methodology, writing (original draft preparation). KN, MB, K-AE, KMA, DOR: conceptualisation, funding acquisition, methodology, supervision, writing (review and editing). HH, GS, AS, NM, AA-L, AF, CN-P, CY, CMC, JK, PB, RP, RDR, ST, SB, UA, ZV: conceptualisation, funding acquisition, methodology, writing (review and editing). LK, KP, MS, MM, NC, SPBHS: conceptualisation, methodology, writing (review and editing). All authors read and approved the manuscript.

**Funding** This work is funded by the Strategic Priority Fund 'Tackling multimorbidity at scale' programme (grant number MR/W014432/1) delivered by the Medical Research Council and the National Institute for Health Research (NIHR) in partnership with the Economic and Social Research Council and in collaboration with the Engineering and Physical Sciences Research Council. The views expressed are those of the author and not necessarily those of the funders, the NIHR or the UK Department of Health and Social Care. The funders had no role in study design, decision to publish or preparation of the manuscript.

**Competing interests** None declared.

**Patient and public involvement** Patients and/or the public were involved in the design, or conduct, or reporting, or dissemination plans of this research. Refer to the Methods section for further details.

**Patient consent for publication** Not applicable.

**Provenance and peer review** Not commissioned; externally peer reviewed.

**ORCID iDs**
Siang Ing Lee http://orcid.org/0000-0002-2332-5452
Holly Hope http://orcid.org/0000-0002-4834-6719
Dermot O'Reilly http://orcid.org/0000-0002-9181-0652
Lisa Kent http://orcid.org/0000-0002-8882-0526
Gillian Santorelli http://orcid.org/0000-0003-0427-1783
Anuradhaa Subramanian http://orcid.org/0000-0001-8875-7363
Amaya Azcoaga-Lorenzo http://orcid.org/0000-0003-3307-878X
Adeniyi Francis Fagbamigbe http://orcid.org/0000-0001-9184-8258
Catherine Nelson-Piercy http://orcid.org/0000-0001-9311-1196
Christopher Yau http://orcid.org/0000-0001-7615-8523
Colin McCowan http://orcid.org/0000-0002-9466-833X
Jonathan Ian Kennedy http://orcid.org/0000-0002-1122-6502
Katherine Phillips http://orcid.org/0000-0003-0674-605X
Megha Singh http://orcid.org/0000-0003-3680-7124
Mohamed Mhereeg http://orcid.org/0000-0003-1241-9549
Neil Cockburn http://orcid.org/0000-0001-9284-6991
Peter Brocklehurst http://orcid.org/0000-0002-9950-6751
Rachel Plachcinski http://orcid.org/0000-0001-9908-0773
Richard D Riley http://orcid.org/0000-0001-8699-0735
Shakila Thangaratinam http://orcid.org/0000-0002-4254-460X
Sinead Brophy http://orcid.org/0000-0001-7417-2858
Utkarsh Agrawal http://orcid.org/0000-0001-5181-6120
Zoe Vowles http://orcid.org/0000-0001-6989-2180
Kathryn Mary Abel http://orcid.org/0000-0003-3538-8896
Krishnarajah Nirantharakumar http://orcid.org/0000-0002-6816-1279
Mairead Black http://orcid.org/0000-0002-6841-8601
Kelly-Ann Eastwood http://orcid.org/0000-0003-3689-0490

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
