## [Reviewer comments · BMJ Open]

ARTICLE DETAILS

TITLE (PROVISIONAL)	Maternal and child outcomes for pregnant women with pre-existing multiple long-term conditions: protocol for an observational study in the United Kingdom
AUTHORS	Lee, Siang Ing; Hope, Holly; O'Reilly, Dermot; Kent, Lisa; Santorelli, Gillian; Subramanian, Anuradha; Moss, Ngawai; Azcoaga-Lorenzo, Amaya; Fagbamigbe, Adeniyi; Nelson-Piercy, Catherine; Yau, Christopher; McCowan, Colin; Kennedy, Jonathan; Phillips, Katherine; Singh, Megha; Mhereeg, Mohamed; Cockburn, Neil; Brocklehurst, Peter; Plachcinski, Rachel; Riley, Richard; Thangaratinam, Shakila; Brophy, Sinead; Hemali Sudasinghe, Sudasing Pathirannehelage Buddhika; Agrawal, Utkarsh; Vowles, Zoe; Abel, Kathryn; Nirantharakumar, Krishnarajah; Black, Mairead; Eastwood, Kelly-Ann; -, on behalf of MuM-PreDiCT

VERSION 1 – REVIEW

REVIEWER	Brown, Hillary University of Toronto at Scarborough, Health & Society
REVIEW RETURNED	28-Oct-2022

GENERAL COMMENTS	This is an interesting protocol of a study aiming to examine maternal and child outcomes in women with multiple long-term conditions compared to those without multiple long-term conditions. The study will use health administrative data from four datasets in the UK. This is an important study that will add to the literature on maternal chronic disease and perinatal and child outcomes. Although I felt the topic was extremely important, I felt the protocol was lacking several major necessary details. Please see below for my specific comments: In the abstract, it was unclear to me what 'women without multiple long-term conditions' includes – i.e., those with 0 and 1 conditions? Why not look at women with 2+, 1, and 0 conditions separately? I felt more information was needed on the comparability of the datasets. For example, the CPRD only includes 5% of UK general practices, with linkages to birth health records, whereas the SAIL includes data 80% of Welsh general practices. Similarly, only the SMR datasets seemed to contain any information on ED visits and hospital admissions (including psychiatric admissions). Finally, the NIMATS and Born in Bradford study appear to be more of a birth cohort created specifically to examine women and children. In other words, these datasets seem to be quite different, and I wonder about the validity of combining them in a meta-analysis.
---

	Why could the authors not conduct their study in one of the datasets only? I would have liked to see more information on the exposure. What conditions are included? Prior multimorbidity studies show a vast difference in the prevalence of multimorbidity based on the definition used. Has this definition been validated? Is it commonly used? Related to this, how will the authors account for the severity of individual conditions? Someone with mild asthma and mild arthritis would have a very different birth experience than someone with severe chronic hypertension and diabetes that is out of control. Yet, in this framework, they would be “weighted” the same (as having 2 conditions). This seems to me to be a very important limitation. How will the ‘different combinations of long-term conditions’ be chosen? Will they be chosen based on clinical relevance or based on the frequency occurring in the population? I wonder whether the authors will be able to completely capture miscarriage in their datasets, since many miscarriages occur at home and do not come to the attention of the health care system. I recommend removing this outcome or reframing it as ‘recognized miscarriage’. Why do antenatal outcomes not include health care use if this is included as a postnatal outcome? I felt the authors could do a better job of conceptualizing the postnatal and long-term outcomes since they are vast (e.g., incontinence and also general health care utilization). As well, it is unclear how long children will be followed for examination of neurodevelopmental disorders, nor what these disorders will include. Mental illness is a part of the multimorbidity definition and is also included as an outcome. This therefore seems a bit tautological. Perhaps this outcome could be re-framed as antenatal and postnatal health care utilization. Otherwise, the authors might want to restrict this analysis to people with multimorbidity based on physical conditions only. I disagree that hierarchical control for confounders allows the authors to assess mediation. This approach will only allow the authors to look at the extent to which a covariate attenuates the main effect for the association; it will not provide information on the magnitude of the effect acting through that mediator. If the authors are interested in mediation, I would suggest using formal mediation analyses. Otherwise, covariates should be limited to confounders. Why is BMI included as a covariate rather than part of the exposure? Many multimorbidity studies now conceptualize BMI as part of the definition of multimorbidity. What will the comparison group be for the analyses looking at different patterns of multimorbidity?
--	---

	I would encourage the authors to reconsider the use of logistic regression to examine the outcomes since many are common and will violate the rare outcome assumption necessary for logistic regression. They may want to consider log binomial regression or modified Poisson regression – both appropriate for directly estimating relative risks. More details are needed on the meta-analysis. What types of models will be used (e.g., random vs. fixed effects) and using what approach? How will the authors ensure consistency in the definitions of the data across datasets? Will a common study protocol be created? How will adherence to the protocol be checked? More details are needed here. I would recommend doing sensitivity analyses in primiparous women given that a ‘chronic condition’ such as heart disease may actually be a consequence of or part of severe maternal morbidity in a previous pregnancy. I think that the clinical implications section needs more thought. The authors state that no clinical guidelines exist for women with multimorbidity in the UK. Although I agree this is a problem, I also wonder what such guidelines would look like, given the vastly different patterns of multimorbidity women might exhibit, and the implications these differences have for clinical care. E.g., a woman with asthma and arthritis will have very different care than a woman with heart disease and diabetes.
--	--

REVIEWER	Dubey, Pallavi Texas Tech University Health Sciences Center El Paso, Ob-Gyn
REVIEW RETURNED	14-Dec-2022

GENERAL COMMENTS	Overall comments: The draft has been written in a hurry and doesn't confirm the BMJ standards. There are multiple grammatical mistakes in the draft and the language is unclear. Line 130: Maternal preexisting multiple long-term conditions have not been defined or explained anywhere in the introduction. The introduction focuses more on the general demographic data rather than being specific on the topic. Methods and Analysis Line 136 Category of observational study should be mentioned here. (E.g. cross sectional, case-control, case-crossover, etc.) Line 150 the information about the database sources is very vast and often unnecessary. The information should be concise and pertain to the study. I would suggest including 5-6 lined paragraphs which should include data information, algorithm, and included population. Line 203: “Exposed group will consist of pregnant women with multiple long-term conditions”. The authors should mention the variables or at least categorize the conditions they plan to include as inclusion criteria. Without the mention of the above-said conditions, this looks very vague and pointless. Line 260: “Children’s mental health and behavioral disorders will also be considered”. How are the authors planning to execute this, since this is an observational study based on data available? How would they link the mother and the child’s MRN? What are the variables they propose to study in children? Line 266: Maternal age. Since the long-term conditions the authors propose to work on have never been mentioned, the covariates association cannot be ascertained at this time by looking at the
---

	protocol. For example, women diagnosed with PCOS and delivering a child at 35 will have a different outcome than a woman diagnosed with chronic hypertension and delivering at the same age. Also, what about co-morbid conditions? A woman diagnosed with PCOS, preeclampsia, gestational diabetes, and acanthosis at the time of delivery, even at a younger age is more prone to adverse delivery outcomes than women only diagnosed with gestational diabetes. The authors have not accounted for the co-morbidities. Line 276: The ethnicity of the mothers has not been defined properly. Will also depend on the conditions they are looking at.
--	---

VERSION 1 – AUTHOR RESPONSE

No	Reviewer's comments	Author's responses
	Editor	
1	Please change the main text heading 'Dissemination' to 'ETHICS AND DISSEMINATION' and move all the information currently under the headings 'Ethics approval' and 'Consent for publication' at the end of the document into this revised section, so these sections are part of the main text.	The 'Ethics approval' and 'Consent for publication' sections have been moved to the main text's 'Dissemination' section, and the subheading has been changed to 'Ethics and dissemination'.
2	Please delete the 'Conclusion' section at the end of the main text – Conclusion sections are not part of our protocol manuscript format.	The 'Conclusion' section has been deleted from the main text.
3	In an appropriate place in the main text of the manuscript, please add some text to indicate the status of the analysis and the planned or expected timeline for its completion.	We have added the following sentence at the start of 'Statistical analyses' section: "We anticipate analyses will commence in June 2023 with study completion by June 2024."
4	Relatedly, in the abstract and main text you indicate that the study will include pregnancies from 2000 to 2019, but please further clarify the timepoint(s) up to which follow-up data will be included in the analysis.	We have added this to the 'Abstract: Methods' section: "Pregnant women aged 15 to 49 years old with a conception date between 2000 and

		2019 in the UK will be included with follow up till 2019." We have also added this to the 'Methods: Outcome' section: "Outcomes will be ascertained from the study datasets (2000-2019) using clinical codes, such as Read, ICD-10 and OPCS codes."
5	Please slightly modify/simplify the article title as follows: "Maternal and child outcomes for pregnant women with pre-existing multiple long-term conditions: protocol for an observational study in the United Kingdom".	The title has been changed as advised to: "Maternal and child outcomes for pregnant women with pre-existing multiple long-term conditions: protocol for an observational study in the United Kingdom".
6	In the 'Methods and analysis' section of the abstract, please join the three paragraphs together into one (as each section should be a single paragraph).	In the abstract, the 'Methods and analysis' section has been merged into one paragraph.
7	Please change the headings 'Availability of data and materials' and 'Author contributions' to 'Data availability statement' and 'Contributors', respectively. Please also delete the Acknowledgements statement, since it is redundant.	The subheadings changes have been made, 'Acknowledgement' has been deleted.
	Reviewer 1	
8	In the abstract, it was unclear to me what 'women without multiple long-term conditions' includes – i.e., those with 0 and 1 conditions? Why not look at women with 2+, 1, and 0 conditions separately?	The main exposure of interest is multiple long-term conditions which we have defined as 2 or more, therefore, the comparator group would be women without multiple long-term conditions (i.e, 0 or 1 long-term conditions). Although this was stated in the 'Comparator' section, it was not clear in the 'Abstract'. We have therefore added the following clarification to the 'Abstract: Background' section: "This observational study aims to compare maternal and child outcomes for pregnant

		women with multiple long-term conditions to those without multiple long-term conditions (0 or 1 long-term conditions).” However, we will also examine the biological gradient by comparing women with increasing counts of long-term conditions, which will include women with 1 long-term condition.
9	I felt more information was needed on the comparability of the datasets. For example, the CPRD only includes 5% of UK general practices, with linkages to birth health records, whereas the SAIL includes data 80% of Welsh general practices. Similarly, only the SMR datasets seemed to contain any information on ED visits and hospital admissions (including psychiatric admissions). Finally, the NIMATS and Born in Bradford study appear to be more of a birth cohort created specifically to examine women and children. In other words, these datasets seem to be quite different, and I wonder about the validity of combining them in a meta-analysis. Why could the authors not conduct their study in one of the datasets only?	The information on the data sources have now been summarised in Table 1: Summary of data sources, with additional commentary on the relative strength of each database. The following has been added to the Discussion section to add reasoning why the five datasets will be used: “Observing similar effect sizes across the different datasets will also increase the confidence in the study findings. Conversely, discrepancy in findings will stimulate further exploration of the datasets which may generate new knowledge.” Meta-analysis will be conducted by primary and secondary care database respectively depending on the data source for exposure and outcomes. This has been added to the Methods section: “The effect sizes will be pooled using random-effects meta-analyses with inverse variance weighting for the primary care and secondary care datasets respectively.” The column headings for Table 1 are: Name of data source, country, population: pregnant women, exposure: maternal multimorbidity status, outcomes: pregnant women, outcomes: children.

		Commentary on the relative strength of each data source: “Table 1 presents the five data sources that will be used. Each UK devolved nation is represented by a population based routine health record datasets, with good national coverage for Wales, Scotland and Northern Ireland and a representative sample for England. The exposure status will be determined from primary care records for Clinical Practice Research Datalink (CPRD) and Secure Anonymised Information Linkage (SAIL), with CPRD GOLD representing currently 5% of UK general practices, and SAIL covering 80% of Welsh general practices. For Scotland’s linked routine records and Northern Ireland Maternity System (NIMATS), the exposure status will be determined from hospital and prescribing records. CPRD and SAIL’s primary care data offer the opportunity to study outcomes that may not be captured in secondary care. For instance, vomiting in pregnancy, miscarriage and neurodevelopmental conditions in children. The Scottish datasets provides detailed information on the different types of hospital attendance, including psychiatric admissions and accident and emergency attendance. NIMATS’s unique first antenatal visit dataset is a good source of pre-pregnancy clinical data not available in other datasets. As routine health records were not collected for research purposes, it is prone to missing data. Therefore, we have also included Born in Bradford, a regional birth cohort (2007-2011) where data were collected systematically and longitudinally from pregnancy, childhood through to adult life.”
10	I would have liked to see more information on the exposure. What conditions are included? Prior multimorbidity studies show a vast difference in the prevalence of multimorbidity	The following sentences have been added to the 'Methods: Exposure' section. We have also added Table 2 which lists the 79

	based on the definition used. Has this definition been validated? Is it commonly used?	conditions included to define the multiple long-term conditions exposure. “Measurements of multiple long-term conditions are variable in existing literature. Currently only Bateman et al’s Maternal Comorbidity Index has been developed specifically for obstetric research. It consists of 20 health conditions and included conditions arising in pregnancy such as gestational hypertension, pre-eclampsia and placenta praevia. This limits the ability to study the impact of pre-existing long-term conditions on maternal and child health and the implication for long-term condition management preconception. In this study, we shall define multiple long-term conditions as two or more long-term physical or mental health conditions that pre-existed before pregnancy. Pregnancy related complications will not be included as they will be studied as outcomes. Multiple long-term conditions will be defined from a list of 79 health conditions previously described in our epidemiological work (Table 2) and will be measured with simple count. This list was compiled from existing multimorbidity literature and a workshop with our multidisciplinary research advisory group, including patient representatives and clinicians. Selection of health conditions were based on: (i) prevalence; (ii) potential to impact on pregnancy outcomes; (iii) considered important by women; and (iv) recorded in the study datasets.”
11	Related to this, how will the authors account for the severity of individual conditions? Someone with mild asthma and mild arthritis would have a very different birth experience than someone with severe chronic hypertension and diabetes that is out of control. Yet, in this framework, they would be “weighted” the same (as having 2 conditions). This seems to me to be a very important limitation.	We recognise this limitation and have attempted to control for the mild conditions with additional criteria such as needing medications. The following is added to the ‘Methods’ section: “The phenome definitions for these health conditions have previously been described in our epidemiological work. For health

		conditions that are transient and episodic in nature (e.g. asthma, eczema, depression and anxiety), we will only include the condition if it is active, which we have defined as requiring a doctors' consultation or medical prescription in the 12 months preceding pregnancy." We have also added the following to the 'Discussion: Limitation' section: "Maternal multimorbidity will be quantified with simple counts. The severity of each health conditions will not be captured and the dose-response relationship will only be reflected in the total number of pre-existing long-term conditions. A systematic review of comorbidity indices used in maternal health research found three indices: Maternal Comorbidity Index, Charlson comorbidity index and Elixhauser comorbidity index. Only the Maternal Comorbidity Index was developed from pregnant and postpartum women. It was developed using hospital data with 20 maternal comorbidities but it included pregnancy related complications and factors such as multiple gestation, gestational diabetes, and hypertension disorder of pregnancy. In contrast, the list of health conditions we will use to define maternal pre-existing multimorbidity is more comprehensive and included leading causes of indirect maternal death (e.g. epilepsy) and mental health conditions."
12	How will the 'different combinations of long-term conditions' be chosen? Will they be chosen based on clinical relevance or based on the frequency occurring in the population?	We have now added further information to the 'Methods: Comparator' section: "The selection of which combinations and clusters of long-term conditions to study will be based on how common they are and their clinical relevance, following consultation with patient representatives and clinicians in our research team."

13	I wonder whether the authors will be able to completely capture miscarriage in their datasets, since many miscarriages occur at home and do not come to the attention of the health care system. I recommend removing this outcome or reframing it as 'recognized miscarriage'.	We have added this limitation to the Discussion section: “Exposure and outcome events are only captured in routine health records when the pregnant women have presented to primary or secondary care and therefore the true prevalence and incidence may be underestimated. Health conditions that are managed conservatively in primary care, such as depression, anxiety and miscarriage, may not be captured in secondary care datasets. Events such as termination of pregnancy that occurred outside of the traditional health care settings may also be under-estimated. Similarly, antenatal hospital admission data may not reflect the full burden of additional antenatal appointments or acute care attendances, as care accessed through other routes may not be captured.”
14	Why do antenatal outcomes not include health care use if this is included as a postnatal outcome?	We will study outcomes based on the findings from a core outcome set we developed in collaboration with women with multiple long-term conditions, health care professionals and researchers in this field. This work is still in submission for publication, but provisionally, antenatal and perinatal acute care service utilisation was not included in the final core outcome set. However we agree with the reviewer that antenatal health care use is important and will study antenatal hospital admission based on data availability across the datasets. This has been added to the Methods: Outcomes section. We have also added the following to the Discussion: Strengths and limitations section: “Similarly, antenatal hospital admission data may not reflect the full burden of additional antenatal appointments or acute care attendances, as care accessed through other routes may not be captured.”

15	I felt the authors could do a better job of conceptualizing the postnatal and long-term outcomes since they are vast (e.g., incontinence and also general health care utilization). As well, it is unclear how long children will be followed for examination of neurodevelopmental disorders, nor what these disorders will include.	The outcomes that will be studied will be finalised based on the results of a core outcome set study, here we provided some provisional examples. The following has been added to the 'Postnatal and long-term outcomes' section: “Postnatal outcomes occur in the 42 days after birth, while long-term outcomes are beyond the peripartum and postpartum period. For women this would include functional outcomes such as incontinence. For children, we will use mother baby linked primary and secondary care data to study long-term outcomes such as congenital anomalies, neurodevelopmental conditions (e.g. autism, attention deficit hyperactive disorder and learning difficulty), cerebral palsy, and chronic lung disease. The length of follow up will depend on the availability of data in the routine health records. For example, CPRD has a median follow up of 5 years. We will also examine postpartum readmission for mother and child.” We have also clarified that outcomes will use data from 2000-2019, indicating follow up will be till 2019 where data is available: “Outcomes will be ascertained from the study datasets (1st January 2000-31st December 2019)”
16	Mental illness is a part of the multimorbidity definition and is also included as an outcome. This therefore seems a bit tautological. Perhaps this outcome could be re-framed as antenatal and postnatal health care utilization. Otherwise, the authors might want to restrict this analysis to people with multimorbidity based on physical conditions only.	We have added the following to the 'Outcomes: Mental health' section to make it clear why and how they are included in planned analyses: We will consider both: (i) incident and (ii) recurrent mental health outcomes, where incident means a woman enters the analysis with no prior record of the specific mental health outcome. A perinatal mental health event is indicated by a primary care visit or hospital admission and includes mental health outcomes of concern in the

		antenatal and postnatal period (e.g. depression, psychosis, post-traumatic stress disorder, self-harm and suicide attempts). Comparing the mental health event rates of pregnant women who have and have not got mental health conditions as part of their multiple long-term conditions will allow us to delineate the contribution of mental and physical morbidity to perinatal mental health outcomes. Children’s mental ill health will also be considered (e.g. depression and anxiety). In the section ‘Covariates’, we have added sentences to explain the additional covariates we will use when considering mental health outcomes. For example, in analyses of mental health outcomes there will be additional covariates. For the mother, we will include history of any mental illness, for the child we will include maternal history of any mental and/ or neurodevelopmental conditions.
17	I disagree that hierarchical control for confounders allows the authors to assess mediation. This approach will only allow the authors to look at the extent to which a covariate attenuates the main effect for the association; it will not provide information on the magnitude of the effect acting through that mediator. If the authors are interested in mediation, I would suggest using formal mediation analyses. Otherwise, covariates should be limited to confounders.	We have removed hierarchical control for confounders from the ‘Methods’ section.
18	Why is BMI included as a covariate rather than part of the exposure? Many multimorbidity studies now conceptualize BMI as part of the definition of multimorbidity.	We have added the following to the ‘Discussion: Strengths and Limitations’ section: “Body mass index, which encompasses underweight and obese categories, will be studied as a covariate instead of being counted as part of multimorbidity. There is

		much debate around whether obesity should be considered a disease or a risk factor for other long-term conditions such as cardiometabolic conditions and cancers. What is clear is pre-pregnancy maternal obesity is associated with adverse pregnancy outcome and dedicated care guideline has been established to manage this risk. Studying body mass index as a separate variable will allow us to examine its independent effect and evidence may reaffirm its role as a modifiable risk factor for pregnant women with multiple long-term conditions.”
19	What will the comparison group be for the analyses looking at different patterns of multimorbidity?	Pregnant women with no multiple long-term conditions (0 or 1 conditions) will be the common comparator group.
20	I would encourage the authors to reconsider the use of logistic regression to examine the outcomes since many are common and will violate the rare outcome assumption necessary for logistic regression. They may want to consider log binomial regression or modified Poisson regression – both appropriate for directly estimating relative risks.	We have changed logistic regression to modified Poisson regression in the ‘Methods: Statistical analyses’ section: “Modified Poisson regression will be performed to estimate the relative risks of study outcomes.”
21	More details are needed on the meta-analysis. What types of models will be used (e.g., random vs. fixed effects) and using what approach? How will the authors ensure consistency in the definitions of the data across datasets? Will a common study protocol be created? How will adherence to the protocol be checked? More details are needed here.	We have added more detail as follows in the ‘Methods’ section: “A federated analysis approach will be used as data governance arrangements do not allow pooling of the data across the four nations. Each dataset will be analysed separately following a common study protocol. A common data model will be established and implemented across the dataset, building on our previous work harmonising the phenome definitions for exposure conditions. The effect sizes will be pooled using random-effects meta-analyses with inverse variance weighting for the primary care and secondary care datasets respectively.”

22	I would recommend doing sensitivity analyses in primiparous women given that a 'chronic condition' such as heart disease may actually be a consequence of or part of severe maternal morbidity in a previous pregnancy.	The proposed additional sensitivity analyses has been added to the 'Sensitivity analyses' section: "We shall conduct sensitivity analyses using (i) complete case analysis, (ii) varying definitions of maternal multiple long-term conditions exposure using D'Arcy et al's core exposure set, and (iii) in primiparous women. The latter is to account for the fact that some long-term conditions can arise from complications from a previous pregnancy. "
23	I think that the clinical implications section needs more thought. The authors state that no clinical guidelines exist for women with multimorbidity in the UK. Although I agree this is a problem, I also wonder what such guidelines would look like, given the vastly different patterns of multimorbidity women might exhibit, and the implications these differences have for clinical care. E.g., a woman with asthma and arthritis will have very different care than a woman with heart disease and diabetes.	The following has been added to the 'Discussion: Clinical implication' section: "The heterogeneity of multiple long-term conditions means an all-encompassing guideline for every possible combination of long-term conditions would not be possible. Indeed the English national guideline for multimorbidity focuses on general approaches such as coordinated and holistic care, improving quality of life by reducing treatment burden and shared decision making between patients and clinicians. A guideline for multiple long-term conditions (multimorbidity) in pregnancy is likely to follow the same principles but with additional focus on the maternity care aspect. The basis of shared decision making is the provision of evidence based information..."
	Reviewer 2	
24	There are multiple grammatical mistakes in the draft and the language is unclear.	We sincerely apologise for this and have now reviewed the manuscript again and rectified the grammatical mistakes.
25	Line 130: Maternal preexisting multiple long-term conditions have not been defined or explained anywhere in the introduction. The introduction focuses more on the general demographic data rather than being specific	Thank you for your comment. We had aimed to highlight lack of research quantifying outcomes for pregnant women with multiple long-term conditions despite its high prevalence, and to build a case why

	on the topic.	routine health records in the UK provides the opportunity to fill this evidence gap. We have now added more information on how maternal pre-existing multiple long-term conditions will be defined in the 'Methods' section and listed the long-term conditions in Table 2. We have also added more content related to multiple long-term conditions in pregnancy to the 'Introduction' section: “Maternal single long-term conditions such as cardiac conditions, chronic kidney disease and epilepsy are associated with adverse pregnancy outcomes. This is likely to be compounded when the pregnant woman has two or more long-term physical or mental health conditions (multimorbidity). Some conditions may need different treatments from different health care teams, thereby increasing the treatment burden and complexity of care. Recent evidence has shown that maternal multiple long-term conditions are associated with adverse outcomes for women and their children, such as severe maternal morbidity and mortality, pre-eclampsia, emergency caesarean birth, preterm birth, and low birth weight. In the UK 2016-18 national maternal mortality report, 90% of women who died during or up to a year after pregnancy had multiple health or social problems. Currently one in five pregnant women multiple long-term conditions prior to pregnancy in the United Kingdom (UK). The number of pregnant women with pre-existing multiple long-term conditions is likely to increase as women are getting pregnant later in life and with higher body weight. As this becomes an increasingly important issue, information on pregnancy, maternal and child outcomes is crucial for women and their health care professionals
--	----------------------	--

		to make informed decisions on preconception and pregnancy care planning. However, there remains a lack of evidence to guide care pathways for pregnant women with multiple long-term conditions.’
26	Line 136 Category of observational study should be mentioned here. (E.g. cross sectional, case-control, case-crossover, etc.)	We have added the following to ‘Study design’: “This is a cohort observational study using data from routine healthcare records and a birth cohort in the UK.”
27	Line 150 the information about the database sources is very vast and often unnecessary. The information should be concise and pertain to the study. I would suggest including 5-6 lined paragraphs which should include data information, algorithm, and included population.	The information on the data sources has now been summarised in Table 1: Summary of data sources , with additional commentary on the relative strength of each database as requested by the other reviewer. The column headings for Table 1 are: Name of data source, country, population: pregnant women, exposure: maternal multiple long-term conditions status, outcomes: pregnant women, outcomes: children. “ Table 1 presents the five data sources that will be used. Each UK devolved nation is represented by a population based routine health record dataset, with good national coverage for Wales, Scotland and Northern Ireland and a representative sample for England. The exposure status will be determined from primary care records for Clinical Practice Research Datalink (CPRD) and Secure Anonymised Information Linkage (SAIL), with CPRD GOLD representing 5% of UK general practices, and SAIL covering 80% of Welsh general practices. For Scotland’s linked routine records and Northern Ireland Maternity System (NIMATS), the exposure status will

		be determined from hospital records and prescribing records. CPRD and SAIL's primary care data offer the opportunity to study outcomes that may not be captured in secondary care. For instance, vomiting in pregnancy, miscarriage and neurodevelopmental conditions in children. The Scottish dataset provides detailed information on the different types of hospital attendances, including psychiatric admissions and accident and emergency attendances. NIMATS's unique first antenatal visit dataset is a good source of pre-pregnancy clinical data not available in other datasets. As routine health records were not collected for research purposes, it is prone to missing data. Therefore, we have also included Born in Bradford, a regional birth cohort (2007-2011) where data were collected systematically and longitudinally from pregnancy, childhood through to adult life."
28	Line 203: "Exposed group will consist of pregnant women with multiple long-term conditions". The authors should mention the variables or at least categorize the conditions they plan to include as inclusion criteria. Without the mention of the above-said conditions, this looks very vague and pointless.	We have now added Table 2 which lists the 79 health conditions we will use to define multiple long-term conditions (multimorbidity) in pregnancy for the proposed study. We have also added further details in the Methods: Exposure section.
29	Line 260: "Children's mental health and behavioral disorders will also be considered". How are the authors planning to execute this, since this is an observational study based on data available? How would they link the mother and the child's MRN? What are the variables they propose to study in children?	Children's long-term outcomes such as neurodevelopmental conditions, mental illness and behavioural disorders will be measured with mother baby linked primary and secondary care data available in the research dataset. More information of the data source has been added to Table 1. We have added the following sentence in the 'Postnatal and long-term outcomes' section:

		“For children, we will use mother baby linked primary and secondary care data to study long-term outcomes such as congenital anomalies, neurodevelopmental conditions (e.g. autism, attention deficit hyperactive disorder and learning difficulty), cerebral palsy, and chronic lung disease. The length of follow up will depend on the availability of data in the routine health records. For example, CPRD has a median follow up of 5 years.”
30	Line 266: Maternal age. Since the long-term conditions the authors propose to work on have never been mentioned, the covariates association cannot be ascertained at this time by looking at the protocol. For example, women diagnosed with PCOS and delivering a child at 35 will have a different outcome than a woman diagnosed with chronic hypertension and delivering at the same age. Also, what about co-morbid conditions? A woman diagnosed with PCOS, preeclampsia, gestational diabetes, and acanthosis at the time of delivery, even at a younger age is more prone to adverse delivery outcomes than women only diagnosed with gestational diabetes. The authors have not accounted for the co-morbidities.	The multiple long-term conditions (multimorbidity) exposure includes a comprehensive list of comorbidities and is now listed in Table 2. We will be studying multimorbidity as a single entity, and examine the dose response relationship by considering increasing total counts of long-term conditions. We have also stated in the ‘Comparator’ section we will be studying the outcomes for different combinations and clusters of multiple long-term conditions.
31	Line 276: The ethnicity of the mothers has not been defined properly. Will also depend on the conditions they are looking at.	We acknowledge that the proposed ethnicity categories of: Asian, Black, Mixed, Other and White, as outlined in the ‘Methods’ section is not at a granular level. This is based on the ethnicity data available and to allow for harmonisation across the study datasets. We have added the following to the Methods: Covariates: Ethnicity section: “Maternal ethnicity will be categorised based on the variables available and to allow for harmonisation across the dataset: Asian, Black, Mixed, Other and White. Where data permits, we may use more granular categories of ethnicity.”

VERSION 2 – REVIEW

REVIEWER	Brown, Hillary University of Toronto at Scarborough, Health & Society
REVIEW RETURNED	13-Jan-2023

GENERAL COMMENTS	Thank you for the opportunity to review this revised manuscript. The authors have been responsive to the suggested changes. I do have a couple of follow-up points: 1. One of the major limitations of this approach is the inability to measure condition severity. The authors mentioned in response to this concern that they will ensure that transient conditions are active in the year prior to pregnancy. However, this does not deal with the issue of severity per se. They do acknowledge in their limitations section that they can only measure simple counts and that other weighted comorbidity indices are inappropriate in their context. However, I think this needs unpacking further – i.e., what could be lost from not being able to measure severity and how this might impact clinical interpretation? 2. The authors state that they will control for BMI and that this will “allow us to examine its independent effect...”. However, typically, the effects of confounders should not be studied, as this leads to “Table 2 fallacy”: https://academic.oup.com/aje/article/177/4/292/147738. I suggest removing this sentence.
---

REVIEWER	Dubey, Pallavi Texas Tech University Health Sciences Center El Paso, Ob-Gyn
REVIEW RETURNED	23-Jan-2023

GENERAL COMMENTS	Thank you for consideration of my comments on your proposed work. I accept all edits done to the manuscript.
--

VERSION 2 – AUTHOR RESPONSE

No	Reviewer’s comments	Author’s responses
	Reviewer 1	
1	One of the major limitations of this approach is the inability to measure condition severity. The authors mentioned in response to this concern that they will ensure that transient conditions are active in the year prior to pregnancy. However, this does not deal with the issue of severity per se. They do acknowledge in their limitations section that they can only measure simple counts and that other weighted comorbidity indices are inappropriate in their context. However, I think this needs unpacking further – i.e., what could be lost from not being able to measure severity and	We have added the following to the limitation section: ‘Nevertheless, when using simple counts to quantify multiple long-term conditions, the severity of each health conditions will not be

	how this might impact clinical interpretation?	captured. The dose-response relationship will only be reflected in the total number of pre-existing long-term conditions. For example, we will not be able to distinguish the outcomes for a pregnant woman with diet controlled diabetes and mild asthma from a pregnant woman with insulin dependent diabetes and brittle asthma. However, pregnant women with severe conditions are more likely to receive intense specialist care than pregnant women with mild conditions. As the number of pregnant women with greater disease severity is likely to be smaller than those with milder condition, adverse pregnancy outcomes may be underestimated.'
2	The authors state that they will control for BMI and that this will “allow us to examine its independent effect...”. However, typically, the effects of confounders should not be studied, as this leads to “Table 2 fallacy”: https://academic.oup.com/aje/article/177/4/292/147738. I suggest removing this sentence.	The following sentence has now been removed: ‘Studying body mass index as a separate variable will allow us to examine its independent effect and evidence may reaffirm its role as a modifiable risk factor for pregnant women with multiple long-term conditions.’